# Stabilization of β-Carotene Liposomes with Chitosan–Lactoferrin Coating System: Vesicle Properties and Anti-Inflammatory In Vitro Studies

**DOI:** 10.3390/foods14060968

**Published:** 2025-03-12

**Authors:** Shuxin Gao, Xiangzhou Yi, Xia Gao, Zhengsen Long, Jingfeng Guo, Guanghua Xia, Xuanri Shen

**Affiliations:** 1Hainan Engineering Research Center of Aquatic Resources Efficient Utilization in South China Sea, Key Laboratory of Seafood Processing of Haikou, School of Food Science and Engineering, Hainan University, Haikou 570228, China; 22210832000029@hainanu.edu.cn (S.G.); gaoxia20202021@163.com (X.G.); longzhengsen@outlook.com (Z.L.); 22220951350074@hainanu.edu.cn (J.G.); xiaguanghua2011@126.com (G.X.); 2School of Food Science and Engineering, Hainan Tropic Ocean University, Sanya 572022, China; xiangzhouyi1995@hainanu.edu.cn

**Keywords:** liposomes, β-carotene, co-modification, stability, hydrogen bond, anti-inflammatory

## Abstract

Liposomes serve as an effective delivery system capable of encapsulating a variety of bioactive substances. However, their structural integrity is susceptible to damage from various environmental factors, which can result in the leakage of the encapsulated bioactive agents. Consequently, identifying effective strategies to enhance the stability of liposomes has become a central focus of contemporary liposome research. Surface modification, achieved by introducing a protective layer on the liposome surface, effectively reduces liposome aggregation and enhances their stability. To this end, we designed a surface modification and constructed liposomes loaded with β-carotene through co-modification with chitosan and lactoferrin, resulting in enhanced stability. This improvement was evident in terms of storage stability, light stability, and in vitro digestion stability. The study investigated the morphology, structure, and physicochemical properties of liposomes with varying degrees of modification. CS-LF co-modified liposomes exhibited significant structural changes, with particle size increasing from 257.9 ± 6.2 nm to 580.5 ± 21.5 nm, and zeta potential shifting from negative to +48.9 ± 1.3 mV. Chitosan and lactoferrin were modified on the liposome surface through electrostatic interactions and hydrogen bonding, forming a dense protective barrier on the lipid membrane. Physicochemical analysis indicated that chitosan–lactoferrin co-modification led to a more ordered arrangement of the phospholipid bilayer, reduced membrane fluidity, and increased membrane rigidity. The interactions between chitosan, lactoferrin, and phospholipids were enhanced through hydrogen bonding, resulting in a denser surface membrane structure. This structural integrity reduced membrane permeability and improved the stability of liposomes under storage conditions, UV irradiation, and in vitro digestion. Additionally, co-modified chitosan–lactoferrin liposomes effectively alleviated lipopolysaccharide-induced inflammatory damage in mouse microglial cells by increasing cellular uptake capacity, thereby enhancing the bioavailability of β-carotene. The results of this study demonstrate that chitosan–lactoferrin co-modification significantly enhances the stability of liposomes and the bioavailability of β-carotene. These findings may contribute to the development of multi-substance co-modified liposome systems, providing a more stable transport mechanism for various compounds.

## 1. Introduction

Liposomes (Lips) are artificial membrane vesicles characterized by hydrophilic and hydrophobic regions, which are separated by a phospholipid bilayer. Due to their biocompatibility, non-toxicity, amphiphilicity, modifiability, and other advantages [1], Lips have been utilized as carrier systems to effectively encapsulate a diverse array of bioactive components, including vitamins, peptides, polyphenols, flavonoids, unsaturated fats, and essential oils [2]. However, the presence of external environmental factors, such as temperature, light, and free radicals, makes Lips susceptible to vesicle aggregation, hydrolysis, and oxidation during storage, which can result in decreased stability [3]. These phenomena may lead to the disruption of the bilayer structure, consequently causing the leakage of bioactive substances. Therefore, identifying suitable methods to enhance the stability of Lips is a critical focus of current Lips research.

One of the core challenges of utilizing liposomes as carriers for active ingredients is their inherent stability issues. Both physical instability (such as aggregation and fusion) and chemical instability (such as phospholipid oxidation and hydrolysis) directly impact the storage life and functional performance of liposomes. In recent years, researchers have made significant strides in enhancing liposome stability through multidimensional innovations, primarily focusing on three key directions: (1) Optimization of lipid composition: Traditional liposomes predominantly use natural phospholipids (such as lecithin) in combination with cholesterol; however, their unsaturated fatty acids are susceptible to oxidative degradation. This has led to a shift in research towards targeted modifications of lipid molecular structures [4]; (2) Surface modification technology: Surface modification of liposomes is a critical approach to enhance their stability, targeting, and functional diversity. This is accomplished by introducing specific molecules or materials to alter the surface characteristics of liposomes [5]; (3) Improvement of preparation processes: Advances in new process technologies, including the use of microfluidic chip technology [6], supercritical CO_2_ methods [7], allow for precise control over the liposome preparation process, thereby enhancing liposome stability.

Among these methods, surface modification effectively reduces the aggregation of liposomes by introducing a protective layer on their surface. This approach enhances liposome stability by influencing particle size, surface charge, and membrane fluidity. Common components employed for the surface modification of liposomes include polysaccharides, phytosterols, proteins, and polyethylene glycol [8,9].

It is worth noting that protein-based surface modification can establish a contact barrier between liposomes and the external environment, thereby enhancing stability [10]. Additionally, this modification facilitates the attachment of specific protein ligands on the surface, promoting the targeting of specific receptors. Compared to other proteins, lactoferrin (LF) not only imparts protective properties to liposomes through surface modification [11], but also significantly enhances cellular uptake efficiency via receptor-mediated endocytosis. This is accomplished through the binding of lactoferrin to low-density lipoprotein receptor-related protein (LRP) or transferrin receptor (TfR) on the cell membrane [12]. Simultaneously, LF-modified liposomes delay the acidification of lysosomes through their surface polysaccharide chains, thereby promoting the release of encapsulated βC into the cytoplasm, which more effectively enhances intracellular utilization.

Furthermore, natural polysaccharides such as pectin and sodium alginate are frequently utilized as stabilizers [13,14]. By modifying polysaccharides, liposomes can maintain structural stability in complex environments and achieve specific biological functions. Among the various polysaccharides used for liposome modification, chitosan (CS), the only cationic polysaccharide found in nature, exhibits excellent biocompatibility and bioadhesion, enabling it to bind to anionic liposomes. Consequently, CS is one of the most commonly employed materials for coating liposomes [15].

Although monolayer modifications can improve the bioavailability of Lips, their effectiveness is still somewhat limited compared to bilayer or multilayer modifications [16]. Multilayer-modified Lips, thanks to their multistructured membranes, demonstrate greater mechanical stability and resistance to leakage [3]. Furthermore, the impact of co-modification with CS and LF on Lips stability remains insufficiently explored, particularly regarding potential non-covalent interactions between the two that may affect the Lips’ surface properties.

Consequently, this study aims to co-modify Lips using CS and LF and investigate their effects on enhancing Lips stability as well as the underlying mechanisms. In this research, Lips were prepared from fish head phospholipids (PLs), and formulations were optimized based on parameters such as particle size, zeta potential, and polydispersity index (PDI). The binding process of CS and LF co-modification to the Lips was analyzed through Fourier-transform infrared (FTIR) spectroscopy and fluorescence spectroscopy. By comparing the storage stability, light stability, and digestive stability of Lips with varying modification levels, we assessed the influence of surface modification on Lips stability. Finally, the anti-inflammatory effects of β-carotene (βC)-loaded Lips were evaluated in lipopolysaccharide (LPS)-induced mouse microglia (BV-2) cells. The findings of this study offer a fresh perspective on enhancing Lips stability and expand the potential applications of Lips in delivering active ingredients.

## 2. Materials and Methods

### 2.1. Materials

Frozen heads of tilapia fish (Oreochromis niloticus) were sourced from Hainan Xiangtai Fisheries Co. Ltd., located in Hainan, China. Cholesterol (≥99%) and βC (≥98%) were acquired from Macklin Reagent in Shanghai, China. The enzyme-linked immunosorbent assay (ELISA) kits were obtained from X-Y Biotechnology, also based in Shanghai, China, while the Cell Proliferation Microplate Assay Kit (MTT) was supplied by Beyotime Biotechnology, again in Shanghai, China. Cell culture materials were provided by Gibco (Thermo Fisher Scientific, Waltham, MA, USA). All additional chemical reagents were of analytical quality and were procured from Sigma-Aldrich in St. Louis, MI, USA, unless specified otherwise.

### 2.2. Extraction and Separation of PLs

PLs were extracted following the method described by Yi et al. [17]. Frozen tilapia heads were first thawed, then homogenized and centrifuged to eliminate blood and excess water. The mixture obtained was subjected to ethanol extraction to produce a fish head paste. For effective dissolution, a mixture of chloroform, methanol, and water (in a ratio of 8:4:3, *v*/*v*/*v*) was employed, followed by centrifugation at 3500× *g* for 10 min at 25 °C to isolate the lower layer. This solution was then combined with 0.9% (*w*/*v*) NaCl at a ratio of 0.2:1 (*v*/*v*), and the supernatant was concentrated through vortexing to retrieve the total lipid content. The separation of various lipids was conducted with a gel medium pressure column purification system. Total lipids underwent solvent elution utilizing chloroform, methanol, and acetone, and the target phospholipids from the tilapia heads were collected by concentrating the methanol eluates, followed by drying with nitrogen and storing at −20 °C. For purity details, please refer to Appendix A.

### 2.3. Lips Preparation

Lips were prepared using the solvent injection method as described by Liu et al. [18], with some modifications. In this procedure, PLs (10 mg) were dispersed with cholesterol (2 mg) in a solvent mixture of ethanol and dichloromethane (2:1, *v*/*v*, 10 mL), and the water bath was heated to 55 °C to ensure complete dissolution. In a separate process, varying amounts of βC (0, 0.005, 0.01, 0.015, and 0.02 g) were completely dissolved in 10 mL of dichloromethane. Subsequently, the dichloromethane solution containing βC was carefully and slowly injected into a 10 mL acetate buffer solution with a concentration of 0.02 mol/L and a pH of 4.0. This buffer solution also included 0.15 mg of Tween-80 as an emulsifying agent. The entire mixture was stirred continuously for 20 min to ensure proper integration and interaction of the components.

The organic solvents were completely removed by rotary distillation (0.06 MPa, 45 °C) to yield a mixed solution (10 mL). The solution was then sonicated (75 W, ice-water immersion) for 30 min to complete the hydration process. The final Lips solution was obtained and stored at 4 °C, protected from light. CS (0, 0.5, 0.6, 0.7, 0.8, and 0.9 g) and LF (0, 0.125, 0.25, 0.375, 0.5, and 1.0 g) were dissolved in an acetic acid buffer solution (0.02 mol/L, 10 mL) and stirred for 2 h to prepare a range of concentrations of CS and LF solutions, respectively. In the modification procedure, the CS solution (10 mL) was slowly dripped into the liposomal suspension (1 mg/mL, 10 mL) at a rate of 0.5 mL/min and stirred to obtain CS-modified lipids. For co-modified Lips, the LF solution was added dropwise at 0.5 mL/min to the CS-modified Lips to form CS-LF-modified Lips. For the two processes of surface modification, the reaction was conducted under stirring conditions at 25 °C and 200 rpm, for 20 min. Lips with varying degrees of modification at each stage were used as the controls. The unloaded βC Lips with different degrees of surface modification were designated as E-lips, E-CS-lips, and E-CS-LF-lips, respectively. Additionally, the loaded βC Lips with varying degrees of surface modification were referred to as β-lips, β-CS-lips, and β-CS-LF-lips, respectively.

### 2.4. Determination of Particle Size and Zeta Potential of Lips

The particle size, zeta potential, and PDI of the Lips were assessed using a Malvern Zeta Sizer Nano ZS-90 analyzer (Malvern Instruments Co., Ltd., Worcestershire, UK) using a dynamic light scattering (DLS) technique. Before the measurements, the Lips were diluted to a concentration of 0.2 mg/mL using an acetate buffer solution (0.02 mol/L, pH 4.0). The Lips underwent a 180 s equilibration period prior to the measurements. The measurements were performed at 25 °C, with three separate readings taken.

### 2.5. Measurement of Encapsulation Efficiency (EE)

The method described by Chen et al. [19] was employed with slight modifications for the determination of EE. One milliliter of Lips solution was vortexed with four milliliters of hexane for 1 min, and the supernatant was subsequently collected. The supernatant was analyzed using a UV-Vis spectrophotometer (TU1901, Purkinje General Instrument Co., Ltd., Beijing, China), and the amount of free βC (*C*_1_) was calculated based on a standard curve. Following this, the lower solution after extraction was mixed with three milliliters of anhydrous ethanol. After completely disrupting the Lips via ultrasonication, four milliliters of n-hexane, followed by mixing and vortexing for 1 min, after which the lower solution was allowed to stand. The absorbance at 455 nm was measured to determine the content of encapsulated βC (*C*_2_). Three parallel experiments were conducted for each group, and a standard curve for βC was established using a series of standard solutions with varying concentrations. The *EE* of Lips was calculated by the following equation:(1)$$\begin{array}{c} EE\left(\%\right)=\frac{C_{1}}{C_{1}+C_{2}}\times 100 \end{array}$$

### 2.6. Scanning Electron Microscopy (SEM)

The microstructure of Lips was observed following the method described by Zhu et al. [20], with minor modifications. Lyophilized Lips samples were placed onto a carrier plate coated with gold for 60 s. A Verios G4 UC field emission scanning electron microscope (FESEM, Thermo Fisher Scientific) was utilized to examine the morphology of the Lips.

### 2.7. Lips Membrane Property Studies

#### 2.7.1. Micropolarity

A solution of pyrene (2 mM, prepared in acetone) was thoroughly combined in a 1:50 (*v*/*v*) ratio with a Lips solution (0.1 mg/mL). This mixture was then incubated for 12 h at 4 °C, protected from light exposure. The excitation wavelength was established at 338.0 nm, while both the excitation and emission slits were adjusted to 5 nm. The spectra were obtained in the wavelength range of 350 to 450 nm at 25 °C, utilizing an F-4700 fluorescence spectrophotometer (Hitachi High-Technologies, Tokyo, Japan). Each sample underwent three measurements. The fluorescence intensities of Lips at 376 nm and 395 nm were labeled I_1_ and I_3_, respectively, with the micropolarity calculated as the ratio of I_1_ to I_3_.

#### 2.7.2. Measurement of Membrane Fluidity

A Laurdan probe was utilized to assess the fluidity of Lip membranes [21]. A solution of the Laurdan probe (1 μmol/L, dissolved in acetone) was combined thoroughly with the Lips solution (0.1 mg/mL) at a volume ratio of 1:50 (*v*/*v*) and then stirred for 20 min in the dark. The assay was performed using an excitation wavelength of 390.0 nm, with both emission and excitation slits adjusted to 5 nm. The spectra were captured between 420 and 600 nm at 25 °C using a Hitachi High-Technologies F-4700 fluorescence spectrophotometer (Tokyo, Japan). Each sample underwent three measurements to confirm accuracy. In this study, the fluorescence emission intensities at 440 nm (representing the gel phase) and 490 nm (indicating the liquid crystalline phase) are denoted as *I*_440_ and *I*_490_, respectively. Each sample was measured three times. The generalized polarization (*GP*) parameter was calculated using the following equation:(2)$$GP=\frac{I_{440}-I_{490}}{I_{440}+I_{490}}$$

### 2.8. FTIR Spectroscopy

The moisture was removed from the samples by freeze-drying. Each sample, weighing 0.5 g, was combined with potassium bromide at a ratio of 1:100 (*w*/*w*), then ground and formed into tablets. FTIR spectra were obtained using a PerkinElmer Frontier infrared spectrometer (Thermo Fisher Scientific, Waltham, MA, USA). The curves were superimposed 64 times, maintaining a resolution of 2 cm^−1^, within a scanning range of 4000 to 400 cm^−1^. Each sample underwent three measurements.

### 2.9. Differential Scanning Calorimetry (DSC)

A DSC131 EVO (Setaram, Lyon, France) device was employed to determine the DSC thermograms of the Lips. The method was modified by Fu et al. [22]. An aluminum sealing pot was used in which 0.2 g of the weighed sample powder was sealed and equilibrated at 25 °C for 10 min. Heating scans were conducted from 25 °C to 150 °C under a nitrogen gas stream, with a set ramp rate of 10 °C/min. An empty aluminum crucible served as the reference. Each test was repeated three times for each sample.

### 2.10. Stability Analysis

#### 2.10.1. Storage Stability

Lips that were freshly prepared were kept at 4 °C for 28 days. During this storage time, the assessments of particle size, zeta potential, PDI, EE, and retention rate (*RR*) of the Lips were conducted at intervals of 0, 7, 14, 21, and 28 days. The retention rate of βC was determined using this equation:(3)$$\begin{array}{c} RR\left(\%\right)=\frac{C_{S}}{C_{0}}\times 100 \end{array}$$
where *C_S_* represents the concentration of βC post-storage and *C*_0_ indicates the initial concentration of βC. Each sample was measured in triplicate.

#### 2.10.2. Photostability

The light stability of Lips was assessed using a modified method by Calligaris et al. [23]. A Lips solution (1 mg/mL, 10 mL) was contained in a transparent glass vial and uniformly placed in a dark room. The ultraviolet (UV) irradiance was set at 30 mW/cm^−2^. The treatment was conducted at 4 °C for durations ranging from 0 to 60 h, with samples collected every 12 h. The UV level, measured using a portable photometer (HD-2102.2 Delta Ohm, Padova, Italy), was approximately 35,000 lx. The control samples were maintained under dark conditions.

#### 2.10.3. In Vitro Digestion Stability

The in vitro digestion stability of Lips was analyzed by preparing simulated gastric fluid (SGF) and simulated intestinal fluid (SIF) according to the ratios established by Brodkorb et al. [24]. During the digestion process, samples were collected every 30 min from the start of mixing to determine the EE and RR of βC. Three replicate tests were conducted for each sample.

##### Simulated Gastric Digestion

The Lips solution (1 mg/mL, 10 mL) was preheated and combined with SGF (7.5 mL, 37 °C), with the pH adjusted to 3.00 ± 0.05 using hydrochloric acid (10 M). Pepsin (2000 Units/mL) was added to the mixture, which was stirred continuously for 2 h at 200 rpm and 37 °C.

##### Simulation of Small Intestine Digestion

The gastric digestion solution was mixed with SIF (11 mL, 37 °C), and 0.4 g of porcine bile extract was added. The pH was adjusted to 7.00 ± 0.05 with sodium hydroxide (10 M). Subsequently, trypsin (800 Units/mL) and lipase (200 Units/mL) were introduced, and the mixture was stirred continuously for 2 h at 200 rpm and 37 °C.

### 2.11. Cell Experiments

LPS induced BV-2 cells were selected as an in vitro model of inflammation for subsequent experiments. BV-2 cultures were grown in Dulbecco’s Modified Eagle Medium (DMEM) supplemented with 10% fetal bovine serum (FBS) and 1% (*w*/*v*) penicillin–streptomycin. The cultures were maintained in an environment of 37 °C and 5% CO_2_.

### 2.12. Cell Viability Assay

Cell viability was assessed following the method established by Patel et al. [25]. BV-2 cells were harvested and plated in 96-well plates at a density of 5 × 10^4^ cells/mL, allowing them to adhere for 24 h. Following this initial incubation, the cells were treated with LPS at a concentration of 100 ng/mL for an additional 24 h. Subsequently, the BV-2 cells were exposed to either Lips (50 μg/mL, diluted in culture medium) or βC (0.5 μg/mL, diluted in culture medium) for another 24 h at 37 °C under an atmosphere of 5% CO_2_. After the specified incubation period, cell viability was assessed using the MTT assay. An MTT solution at a concentration of 5 mg/mL was prepared according to the manufacturer’s guidelines, and 10 μL of this solution was added to each well, followed by a 4 h incubation at 37 °C. Subsequently, 100 μL of formazan solution was introduced into each well, and absorbance readings were taken at 570 nm to determine cell viability.

### 2.13. Measurement of Inflammatory Factor Levels

BV-2 cells were inoculated in 6-well plates at a density of 1 × 10^5^ cells/mL and incubated for 24 h. Following LPS induction and Lips treatment, the cells were collected by centrifugation, and the levels of inflammatory factors (IL-1β, IL-6, IL-10, TNF-α) were determined using an ELISA kit (X-Y Bio-technology, Shanghai, China). Inflammatory factor levels were calculated according to the manufacturer’s instructions.

### 2.14. Statistical Analysis

All measurements were conducted three times. The results were analyzed through one-way ANOVA using the Tukey method for post hoc testing, using SPSS 26.0 software (SPSS Inc., Chicago, IL, USA). Visualization of all data were carried out using Origin 2024 software (Northampton, MA, USA).

## 3. Results and Discussion

### 3.1. Optimization of Lips Formulation

The physicochemical properties of Lips are typically assessed through variations in particle size, zeta potential, PDI, and EE [26]. To determine the optimal modification ratios of the Lips system, we screened the addition of βC, CS, and LF to the Lips. Lips (β-lips) containing varying concentrations (0–2%) of βC were prepared, and the effects of these concentrations on particle size, zeta potential, PDI, and EE were evaluated, as summarized in Table 1. The unloaded βC Lips exhibited a particle size of 167.2 nm and a PDI of 0.28, indicating a homogeneous dispersion. With increasing concentrations of βC, both the particle size and the absolute value of the zeta potential of the Lips increased. This phenomenon aligns with observations made by Ma et al. [27]. The βC, characterized by its rich conjugated double bonds and strong hydrophobicity, was encapsulated within the phospholipid bilayers of the lipids [28], resulting in increased spacing between the phospholipid molecules and consequently larger Lip particle sizes. The EE of the Lips initially increased with the loading of βC, reaching a maximum of 1.0%, after which it gradually decreased. This decline can be attributed to the limited capacity of the phospholipid bilayer to accommodate additional βC. Therefore, 1% βC was selected for subsequent experiments.

Lips were prepared from tilapia head PLs, primarily composed of phosphatidylcholine, which ionizes the Lips surface, resulting in a negative potential state due to its carboxyl group [15]. CS, a cationic polysaccharide, can bind to Lips through electrostatic interactions. To determine the optimal concentration for CS modification, we assessed the effects of varying CS concentrations on Lips. As shown in Table 2, the particle size, zeta potential, and PDI of unmodified Lips were 259.2 nm, −33.0 mV, and 0.27, respectively. Upon CS modification, the Lips potential shifted from negative to positive. Specifically, as the CS concentration increased, the potential changed from −33.0 mV to 51.1 mV, while the particle size also increased during the modification process. These findings confirm the successful modification of the Lips surface with CS. The positively charged amino groups of CS interact with the negatively charged groups (e.g., carboxyl groups) [2] on the Lips surface, resulting in a modification of the Lips potential. At lower CS concentrations (0.5–0.6%), there was a gradual increase in Lip particle size, accompanied by an increase in PDI to 0.55, indicating that the particle size distribution was not uniform under low concentration modifications with CS. This phenomenon may be attributed to incomplete binding of CS to the phospholipid molecules at low concentrations, which is consistent with the phenomenon observed by Guan et al. [29]. When the CS concentration was increased to 0.8%, the Lip particle size rose to 561.6 nm, with a zeta potential of approximately 51.1 mV, and stabilized thereafter. The charged sites of CS continued to bind to the negatively charged sites of Lips until saturation was achieved in the modification process. However, an excess concentration of CS (1.0%) resulted in aggregation among the CS molecules, which also attracted previously modified CS on the Lips surface, leading to the separation and subsequent reduction in system stability. In conclusion, at a CS concentration of 0.8%, the Lips exhibited a stable state characterized by a particle size of 561.6 nm, a zeta potential of 51.1 mV, a PDI of 0.46, and a high βC EE. Therefore, 0.8% CS was selected for subsequent experiments.

The effects of LF modification on Lip particle size, zeta potential, PDI, and EE were further evaluated, and the results are presented in Table 3. The modification of CS on the Lips surface introduced amino groups [15], creating additional potential binding sites for LF. Consequently, the binding of LF to the Lips surface was favored, resulting in the formation of a co-modified Lips system. The incorporation of more LF provided additional carboxyl groups [30], which enhanced the stabilization of the Lips system. As the proportion of LF increased, the Lip particle size initially increased and subsequently stabilized, indicating that LF surface modification reached saturation. The zeta potential and EE exhibited only minor changes with increasing LF proportions. However, a concentration of LF at 0.5 mg/mL resulted in a significant increase in PDI, suggesting that the aggregation of excess LF in the solution adversely affected system stability. Therefore, an LF concentration of 0.375 mg/mL was selected for Lips modification.

The system characterization of Lips with varying degrees of modification, assessed in terms of apparent appearance, particle size, zeta potential, and PDI, is illustrated in Figure 1a,b. Regarding the apparent appearance, the Lips solution transitioned slightly from clarified to turbid, attributed to the incorporation of CS and LF with higher molecular weights, as further evidenced by the PDI results. Overall, the Lips system demonstrated an increase in particle size following surface modification, along with a shift in zeta potential from negative to positive, enhancing stability in absolute value. The detailed material ratios for the final system construction can be found in Appendix A.

### 3.2. Microstructural Observation of Lips

The microstructure of Lips was examined using scanning electron microscopy (SEM). As illustrated in Figure 1c, both E-lip and β-lip exhibited distinct spherical-like characteristics typical of liposomal structures. CS modification led to a notable increase in Lip particle size and an angular morphology, accompanied by polysaccharide attachment [31]. Subsequent modification with LF resulted in the appearance of protein blobs on the surface of the Lips system, indicating successful LF incorporation onto the Lips surface. This phenomenon aligns with the findings reported by Zhang et al. [10]. No significant morphological differences were observed between the loaded and unloaded βC Lips, suggesting that encapsulated βC does not substantially affect the Lips structure [32]. Morphologically, the modified Lip particle size was consistent with the results presented in Figure 1b. These findings demonstrate that the CS and LF modifications were successfully incorporated into the Lips surface, thereby altering the appearance of the Lips.

### 3.3. Lips Membrane Property Studies

#### 3.3.1. Membrane Micropolarity

As an efficient hydrophobic probe, pyrene has been utilized to investigate the micropolarity of bilayers [33]. Pyrene can penetrate the phospholipid bilayer and is instrumental in detecting the micropolar environment within the Lips membrane. The ratio I1/I3 indicates the micropolarity of the Lips membrane, with the measurement results presented in Figure 2a,b. Notably, the micropolarity progressively decreased with the modification of the Lips surface. This decreasing trend was consistently observed in both loaded and unloaded βC Lips. Surface modification of CS results in the formation of a lamellar structure, which may contribute to a reduction in hydrophobicity, consequently leading to a decrease in micropolarity [34]. Additionally, LF undergoes modifications that enhance the density of the surface structure, further decreasing micropolarity. Both CS and LF can adhere to Lips by covering their surfaces and inserting hydrophobic groups into the hydrocarbon chains of the bilayer [35]. The development of a dense surface structure provides a physical barrier that enhances Lips stability.

#### 3.3.2. Membrane Fluidity

Lips mobility can significantly influence drug release and structural stability. Laurdan probes can spontaneously integrate into the bilayer and are sensitive to their surrounding environment [36]. The GP values, calculated from fluorescence intensities at 440 nm (gel phase) and 490 nm (liquid crystal phase), elucidate changes in Lips mobility in relation to the probe. The GP values, along with the fluorescence emission spectra of Laurdan, are presented in Figure 2c and Figure 2d, respectively. Lips membrane fluidity is contingent upon the lipid chain length and the degree of unsaturation of the acyl chains [21]. The Lips were prepared from PLs containing unsaturated acyl groups, resulting in fluid membranes for both E-lips and β-lips. The GP values recorded were −0.070 ± 0.002 and −0.063 ± 0.005, respectively. Surface modification with CS led to an increase in the GP value, indicating that the Lip membranes became stiffer while remaining in a fluid state. Following further modification with LF, the GP values increased significantly; specifically, the β-CS-LF-lips GP values transitioned from negative to positive, indicating a decrease in membrane fluidity and an increase in membrane rigidity. These results may be attributed to the binding of CS to LF during co-modification, which alters the membrane structure of the Lips. The intermolecular interactions resulting from this co-modification tightly bind macromolecules to phospholipid molecules, substantially enhancing the membrane rigidity of the Lips system [17]. The polypeptide chain of LF is rich in hydroxyl/carboxyl groups, while CS is rich in amino/hydroxyl groups that form hydrogen bonds. The carboxyl groups (-COO^−^) of LF and the amino groups (-NH_3_^+^) of CS form multiple hydrogen bonds, increasing the thickness of the modification layer and resulting in enhanced membrane rigidity. Additionally, the hydrophobic domain of lactoferrin may insert into the membrane and bind with the hydrophobic acyl chains of phosphatidylcholine, further restricting segmental motion. This also leads to increased rigidity of the liposome membrane, which is consistent with the experimental results showing reduced membrane fluidity.

### 3.4. FTIR Analysis

The effect of CS and LF on Lips moieties and intermolecular interactions during modification was investigated using FTIR, with the results presented in Figure 3a. In the FTIR spectra of βC, a sharp peak corresponding to the trans conjugated C-H stretching vibration was observed at 965.5 cm^−1^, which was utilized to identify the characteristic peak of βC [37]. Notably, none of the samples after Lips loading (β-lips) exhibited the characteristic peak of βC, indicating that βC has been successfully encapsulated without any observed leakage. A broad peak in the range of 3600–3200 cm^−1^ was detected in all Lips spectra, attributed primarily to N-H and O-H stretching vibrations. Two significant shifts in this peak were observed (from 3433 to 3426 to 3318 cm^−1^) following the successive modifications of CS and LF, accompanied by an increase in the absorption intensity of the O-H stretching vibration. This may be indicative of hydrogen bonding interactions between the Lips and CS and LF, resulting in enhanced N-H and O-H stretching vibrations [38]. The peaks at 2925 cm^−1^ and 2853 cm^−1^ correspond to the stretching vibrations of the methylidene (-CH_2_-) groups in the Lips, while the peak at 1738 cm^−1^ is attributed to the stretching vibration of the carbonyl group (C=O) in the phospholipid of the Lips, and the peak at 1086 cm^−1^ is associated with the symmetric stretching vibration of the phosphodiester group (PO^2−^) [16]. The intensities of these signals varied after modification. In the presence of CS and LF, the absorption of CH_2_ stretching vibrations showed minimal displacement, indicating that the modification process does not significantly affect the hydrocarbon chain of the Lips. This finding aligns with the results reported by Tan et al. [3]. However, the intensity of the C=O stretching vibration of PLs demonstrated a gradual decrease throughout the modification process. This suggests that the formation of electrostatic interactions between the carbonyl and amino groups of the bilayer, CS, and LF contributes to this phenomenon [39]. The proton head of the phospholipid molecule is particularly sensitive to hydrogen bond formation, as evidenced by the peak displacement of PO^2−^ (from 1097 to 1087 cm^−1^) observed during the modification of CS and LF. This displacement confirms the generation of hydrogen bonding structures between the bilayer at the Lips interface and both CS and LF [16]. In conclusion, βC was successfully encapsulated in Lips. Concurrently, electrostatic interactions between the Lips and CS and LF occurred, and the surface modification enhanced the hydrogen bonding among CS, LF, and the Lips. This may represent the primary driving force for the stability of the multilayer modification system.

### 3.5. DSC Analysis

DSC is employed to assess the changes in the thermodynamic parameters during the transition of a substance from an ordered gel state to a disordered liquid crystal state upon continuous heating [13]. In this study, we monitored the phase behavior changes in Lip samples resulting from the addition of CS and LF using DSC, with the results illustrated in Figure 3b. The data indicates that all Lip systems exhibit a broad peak, with the turning point representing the phase transition temperature (T_m_) of the Lips system [40]. This phase transition, characterized by the melting of the hydrocarbon chains in the phospholipid molecules, shows a shift from a fluctuating gel phase to a layered liquid crystal phase. The T_m_ for E-lips and β-lips are 95.4 °C and 88.6 °C, respectively. The rigid structure of βC can diminish the hydrophobic interactions between the carbon chains of PLs, while the intermolecular interactions between βC and Lips contribute to an increase in disorder within the phospholipid bilayers, potentially explaining the observed decrease in T_m_ following the incorporation of βC. The modification of Lips can either elevate or lower the phase transition temperature, contingent upon whether the modifying substance interacts with the acyl chains or polar heads of the PLs within the Lips [41,42]. Notably, the modification with CS results in a gradual increase in T_m_, reaching 101.2 °C for E-CS-lips and 99.2 °C for β-CS-lips. This increase in T_m_ suggests that the CS coating interacts with the polar groups in the PLs, enhancing hydrogen bonding within the Lips system [43]. Furthermore, with the modification of LF, the T_m_ values further rise to 106.8 °C and 107.7 °C, respectively. This phenomenon may be attributed to the increased membrane rigidity resulting from surface modification, which led to a more ordered arrangement of the Lips bilayer and an elevation in the phase transition temperature. This observation aligns with previous findings regarding Lips membrane fluidity [44]. In conclusion, the co-modification of CS and LF significantly enhanced the phase transition temperature of the Lips system. Interactions with the Lip membranes facilitated the formation of surface structures, thereby improving the thermal stability of the Lips.

### 3.6. Stability Analysis

#### 3.6.1. Storage Stability

Lips serve as carriers for unstable active ingredients due to their amphiphilic nature; however, they are also inherently susceptible to aggregation, rupture, oxidation, and degradation, which can result in leakage of the encapsulated material. The shelf life of Lips is contingent upon their stability during storage [45]. Consequently, the storage stability of Lip systems with varying degrees of modification was investigated and evaluated through measurements of particle size and zeta potential, with results presented in Figure 4a,b. After 28 days of storage, all Lips exhibited an increase in particle size, likely due to aggregation and flocculation during this period. Notably, the unmodified Lips experienced a rapid increase in particle size between days 7 and 14, a phenomenon not observed in the CS- and LF-modified groups. The coating formed by CS chains enhanced the stability and environmental tolerance of the Lips. Furthermore, the co-modification with CS and LF created a hydrophilic barrier on the Lips surface [31], which effectively inhibited their aggregation and fusion.

The zeta potential analysis indicated that the absolute value of the Lips potential gradually decreased during storage. This decreasing trend varied among different groups, with the β-CS-LF-lips group exhibiting the slowest rate of decline. This phenomenon is likely attributable to the shielding of surface charge resulting from vesicle aggregation [17]. The reduction in surface charge leads to insufficient repulsion between vesicles, thereby increasing the potential for particle aggregation or sedimentation [46]. The findings suggest that the co-modification of Lips with CS and LF not only enhanced the surface charge of the Lips but also improved their ability to maintain this charge [47]. Chitosan effectively prevents the aggregation of liposomes during a 28-day storage period through an electrostatic stabilization mechanism. The key mechanism is as follows: In a weakly acidic environment, chitosan (CS) molecules have their amino groups (-NH_2_) protonated to form positively charged -NH_3_^+^, which preferentially adsorb onto the negatively charged liposome surface. This significantly increases the liposome’s surface potential from −25.4 mV to 51.1 mV. The strong positive charge layer overcomes the van der Waals attraction between liposomes through charge repulsion, inhibiting particle contact and aggregation. The augmented electrostatic repulsive force mitigated the tendency for particle size increase due to vesicle aggregation, which is advantageous for preserving the structural stability of the vesicles during storage. CS-modified liposomes maintain a high ζ potential of +43 mV, with the particle size only slightly increasing to 46 nm. Additionally, the linear polysaccharide chains of CS further delay membrane fusion through local steric hindrance effects, ensuring long-term dispersion stability and demonstrating excellent storage tolerance in food or drug delivery systems.

βC possesses multiple conjugated double bonds in its structure, rendering it susceptible to degradation by adverse environmental factors such as oxygen and light [48]. As illustrated in Figure 4c, the retention rate of βC increased from 26.5% in β-liposomes to 37.7% in β-CS-LF-liposomes after 28 days of storage at 4 °C. This improvement can be attributed to the various protective mechanisms of the composite modification layer. Firstly, positively charged chitosan binds to the phosphate groups of the phospholipid bilayer through electrostatic attraction, which reduces the lateral mobility of the lipid bilayer and the diffusion coefficient of the cargo [48], effectively inhibiting the migration of βC from the liposome core to the aqueous phase. Secondly, lactoferrin forms a co-modification layer with chitosan through hydrophobic hydrogen bonding, and its dense cross-linked network decreases the fluidity of the lipid bilayer, thereby blocking external oxygen molecules and other factors that could disrupt the liposome system. Concurrently, the glucosamine units of chitosan scavenge reactive oxygen species (ROS) in the system via an electron transfer mechanism [49], and this antioxidant activity reduces the isomerization rate of βC, thereby achieving synergistic protection through both physical barriers and chemical means. Following the co-modification of chitosan and lactoferrin in liposomes, the retention rate (RR) was significantly enhanced.

#### 3.6.2. Photo Stability

The β-carotene molecule is characterized by 11 conjugated double bonds [50], which allow uncoordinated electrons to be easily activated by light, heat, or free radicals, thereby initiating a chain oxidation reaction. In the food industry, protection against this reaction is achieved through methods such as microencapsulation [37] or the addition of antioxidants [51]. Ultraviolet light primarily disrupts the structure of liposomes through photooxidation. Its core mechanisms can be summarized as follows: (1) The energy of ultraviolet light is absorbed by the double bonds of phospholipids, triggering a photosensitive reaction that induces the isomerization of polyunsaturated fatty acids in phospholipids, which leads to increased membrane disorder [52]; (2) The photolysis of water molecules generates reactive oxygen species (ROS) [53], which disrupt the liposome structure through chain reactions, potentially resulting in liposome collapse and leakage of encapsulated substances; (3) Water molecules and secondary products of lipid oxidation, such as 4-hydroxynonenal [54], further damage the liposomes. As analyzed in Figure 5a–c, βC was significantly degraded (83%) within 24 h of UV irradiation. In contrast, the Lip systems encapsulating βC (β-lips, β-CS-lips, β-CS-LF-lips) exhibited degradation rates of only 66%, 53%, and 41%, respectively. Notably, as shown in Figure 5a, β-CS-LF-lips retained 42.5% of βC after 60 h of UV exposure. These results indicate that the modified Lips are more effective at enhancing the resistance of βC to light damage. Furthermore, the co-modification demonstrated a superior effect in increasing the RR of βC, corroborating the findings of Baek et al. [55]. The observed increase in retention can be attributed to the limited interaction of βC, when loaded in Lips, with oxygen and free radicals in the surrounding environment, facilitated by the physical barrier provided by CS and LF [55]. This barrier reduces the mobility of the Lips membrane and preserves its structural integrity. Concurrently, the complex surface structure limits the mobility of free radicals and attenuates the efficiency of photo-oxidation [56]. CS contributes amino groups that enhance hydrogen bonding forces with Lips, thereby improving membrane strength and subsequently reducing βC degradation. Secondly, both CS, which is rich in amino [15] and hydroxyl groups, and the chelating effect of LF [57] can exert antioxidant roles, mitigating the damage to βC caused by UV light.

#### 3.6.3. Digestive Stability

To further evaluate the stability of Lips during gastrointestinal digestion, in vitro digestion simulation experiments were designed to investigate the behavior of Lips with varying degrees of encapsulation in releasing βC. The results are presented in Figure 6. After 120 min of digestion in the SGF stage, only minor changes were observed in the EE of β-lips, β-CS-lips, and β-CS-LF-lips, with decreases of 10.3%, 8.4%, and 7.3%, respectively. Overall, the digestive stability of all Lips during the SGF phase was relatively high; however, the co-modification of CS and LF significantly enhanced the stability of the Lips in SGF. This improvement may be attributed to several factors: (1) The absence of suitable enzymes for degrading PLs in SGF allowed the phospholipid membrane of the Lips to remain intact, effectively retaining the encapsulated βC [15]. (2) A surface barrier formed by the coating inhibited the further release of βC from the Lip membranes. (3) The structure of a single CS-modified layer may diffuse or solubilize in acidic media [16]. Continued co-modification with LF strengthened the Lips-modified layer through the formation of hydrogen bonds, thereby enhancing Lips stability in SGF. To visually demonstrate the release process of βC, Figure 6b illustrates the release profile of βC during simulated digestion. The release curves for the three Lip samples are closely aligned in SGF, further indicating that the Lips remain stable under acidic conditions and in the absence of suitable digestive enzymes.

Upon entering the SIF phase, the degradation rate of the Lips increased significantly, leading to the substantial release of βC. Within 30 min, the EE of β-lips, β-CS-lips, and β-CS-LF-lips decreased by 18%, 12.6%, and 9%, respectively. Subsequently, the release rates of βC reached 86.5%, 61.8%, and 55.0% at 120 min of SIF digestion. The slopes of the release curves indicated that β-lips were released more rapidly in the intestine, primarily due to the swift hydrolysis of lipids facilitated by the combined action of tryptic enzymes and bile salts present in the SIF, which directly led to the collapse and destruction of the Lips structure, resulting in the leakage of βC [17,58]. The release of Lips-loaded βC was slowed after co-modification with CS and LF. The binding of CS and LF to bile salts and digestive enzymes can effectively reduce the damage to Lips during digestion. Pancreatic lipase, aided by bile salts, exposes its active site, specifically hydrolyzing the ester bonds of phospholipids and further disrupting the lipid framework [59]. During the co-modification process, CS adsorbs onto the surface of liposomes through cationic electrostatic interactions, forming a protective ‘shell-like layer’ that inhibits the direct insertion of bile salts. LF further enhances this physical barrier due to its multi-domain characteristics. Previous micro-polarity measurements have shown that the hydrophobicity of liposomes gradually decreases with surface modification, indicating that the co-modified liposomes shield the hydrophobic sites. This demonstrates that the co-modified liposomes obstruct the invasion of bile salts, reduce the binding sites for pancreatic lipase, and enhance the stability of the liposomes during digestion. The release of βC from SIF primarily depends on membrane permeability, which is influenced by the fluidity of the lipid bilayer [16]. The slower release from CS Lips may result from the interaction of CS molecules with the Lips surface, forming a modified shell structure that is further reinforced by LF modification. Consequently, the fluidity of the Lips membrane is more restricted, and permeability is reduced. Therefore, the co-modification of Lips with CS and LF is advantageous for improving the stability of Lips in the gastrointestinal tract and delaying the release of βC, which will enhance the circulation time and bioavailability of Lips in vivo.

### 3.7. Study of Anti-Inflammatory Properties of Liposomal System

βC is a natural pigment commonly found in nature, characterized by its molecular structure, which contains several conjugated double bonds. Research has demonstrated that βC exhibits anti-inflammatory effects by regulating cytokines [60] and inhibiting inflammatory signals [61]. However, lower bioavailability significantly limits its anti-inflammatory potential. Co-modified Lips composed of CS and LF have proven more effective in enhancing the stability and digestive properties of βC, potentially leading to a more pronounced anti-inflammatory effect and increased activity.

LPS, a major component of the outer membrane of Gram-negative bacteria, triggers the production of inflammatory factors and is commonly employed to mimic and study inflammatory responses [62]. BV-2 cells, a mouse microglial cell line with macrophage-like functions, are capable of producing a variety of inflammatory factors in response to infection and inflammatory stimuli, making them widely utilized in inflammation-related research [63]. Consequently, LPS-induced BV-2 cells were selected as a model for inflammation to investigate the ameliorative effects of βC-loaded Lips on inflammatory processes.

#### 3.7.1. The Effect of Lips on LPS-Induced BV-2 Cell Viability

The cell viability among different samples following LPS treatment is illustrated in Figure 7a. The viability of BV-2 cells post-LPS treatment was significantly reduced, which was attributed to apoptosis induced by inflammatory factors produced during LPS induction. However, after treatment with βC and Lips, cell viability was significantly restored. Notably, the highest cell viability was observed in the β-CS-LF-lips group, indicating that βC enhanced the bioactivity of βC utilization following co-modification with Lips loading. This trend is consistent with the findings from the previous experimental stability assay. Following the co-modification of CS with LF Lips, the loaded βC was better protected and exhibited greater bioactivity. Consequently, β-CS-LF-lips are more effective in reducing the levels of inflammatory factors and mitigating the damage caused by these factors to cells [64]. In summary, Lips serve as effective carriers of βC, enhancing cell viability. Additionally, the stability of Lips can be improved through surface modifications with CS and LF, thereby enhancing the utilization of βC and more effectively reducing the inflammatory response in BV-2 cells.

#### 3.7.2. The Effect of Lips on LPS-Induced Inflammatory Factor Secretion in BV-2 Cells

To further investigate the ameliorative effect of Lips loaded with βC on inflammation, we measured the secretion levels of inflammatory factors in BV-2 cells following Lips treatment. The factors assessed included the pro-inflammatory cytokines IL-6, TNF-α, and IL-1β, as well as the anti-inflammatory cytokine IL-10. Pro-inflammatory cytokines are a class of proteins capable of stimulating, recruiting, and expanding immune cells [65], thereby promoting inflammation. In contrast, anti-inflammatory factors, secreted by macrophages in response to stimulation, serve to attenuate or inhibit the inflammatory response, preventing excessive inflammation [66]. In the central nervous system (CNS), these cytokines interact synergistically and play a crucial role in regulating inflammation, immune response, and microglial activation [63,67,68].

As shown in Figure 7b–d, after LPS treatment, the concentrations of TNF-α, IL-1β, and IL-6 were significantly elevated (*p* < 0.05) compared to the unstimulated control group, indicating that BV-2 cells exhibit a pro-inflammatory polarization response to LPS stimulation [69]. Treatment with the positive control dexamethasone (DX) effectively suppressed the cellular inflammatory response, leading to a significant reduction in the levels of pro-inflammatory factors. Additionally, after treatment with various modified Lips that were unloaded with βC, all pro-inflammatory factors exhibited a slight decrease compared to the blank group, suggesting that the Lips system itself may possess some anti-inflammatory properties. The secretion levels of pro-inflammatory factors (TNF-α, IL-1β, and IL-6) were significantly decreased in βC, as well as following treatment with loaded βC Lips. This indicates that βC exerts substantial anti-inflammatory activity compared to E-lips. Regarding IL-1β and IL-6, the β-CS-LF-lips group demonstrated extremely low secretion levels of these pro-inflammatory factors. This suggests that co-modification with CS-LF enhances the stability of βC, thereby improving its anti-inflammatory effects and effectively increasing its bioavailability. As illustrated in Figure 7e, the level of the anti-inflammatory factor IL-10 was not significantly altered by LPS treatment. Conversely, the expression level of IL-10 was significantly elevated following various Lips treatments. These findings imply that βC and Lips treatment may effectively mitigate inflammation-induced cellular damage by inhibiting the expression of inflammatory factors while promoting the expression of anti-inflammatory factors. Similar results were reported by Lang et al. [67]. Additionally, Lactoferrin significantly improves the cellular uptake efficiency of liposomes via receptor-mediated endocytosis. This occurs through the binding of lactoferrin to a low-density lipoprotein receptor-related protein (LRP) or transferrin receptor (TfR) present on the cell membrane [12]. Lu et al. [11] discovered a similar phenomenon, where LF modification can enhance the cellular uptake of nanoparticles. This aligns with the trends observed in our results. Meanwhile, LF-modified liposomes delay lysosomal acidification through their surface polysaccharide chains [70], which promotes the release of the encapsulated βC into the cytoplasm, thereby enhancing intracellular utilization more effectively. In summary, the co-modification of CS and LF with Lips loading resulted in increased stability of βC and enhanced anti-inflammatory effects. Furthermore, LF modification improves Lip uptake in BV-2 cells, addressing the issue of low bioavailability of βC.

## 4. Conclusions

In summary, this study successfully utilized CS and LF for the surface modification of loaded βC Lips. The physicochemical properties and structural states of the Lips system were investigated under varying degrees of modification. After co-modification with chitosan and lactoferrin (referred to as β-CS-LF-lips), the particle size of the liposomes significantly increased from 257.9 nm to 580.5 nm, while the zeta potential changed from negative to positive, reaching +48.9 mV. Membrane property studies revealed that the interaction between CS-LF co-modification and the Lips membrane led to reduced membrane fluidity and increased membrane rigidity. FTIR analysis indicated that the stability of the system was enhanced through the formation of additional hydrogen bonds between CS, LF, and the phospholipid bilayer. DSC analysis showed that CS-LF co-modification resulted in a more ordered distribution of phospholipids, increasing the Tm value of Lips (88.6–107.7 °C) and enhancing their resistance to thermal instability. Stability analysis demonstrated that the CS-LF co-modification formed a hydrophilic barrier on the Lips surface, thereby enhancing surface charge repulsion. By restricting free radical mobility and reducing the accessibility of digestive enzymes, the stability of Lips under UV irradiation and during digestion was improved, while also extending the storage time of Lips (the retention rate of βC increased from 26.5% in β-liposomes to 37.7% in β-CS-LF-liposomes after 28 days). Moreover, the use of CS-LF co-modified Lips reduced the levels of the pro-inflammatory factors (TNF-α, IL-1β, and IL-6) and increased the levels of anti-inflammatory factors (IL-10), thereby enhancing the bioactivity of βC. In conclusion, the co-modification of encapsulated βC Lips surfaces with CS and LF is an effective strategy for improving the stability of Lips. This study provides new insights into the construction of multi-substance co-modification systems to enhance Lips stability and further expand their applications in food.

## Figures and Tables

**Figure 1 foods-14-00968-f001:**
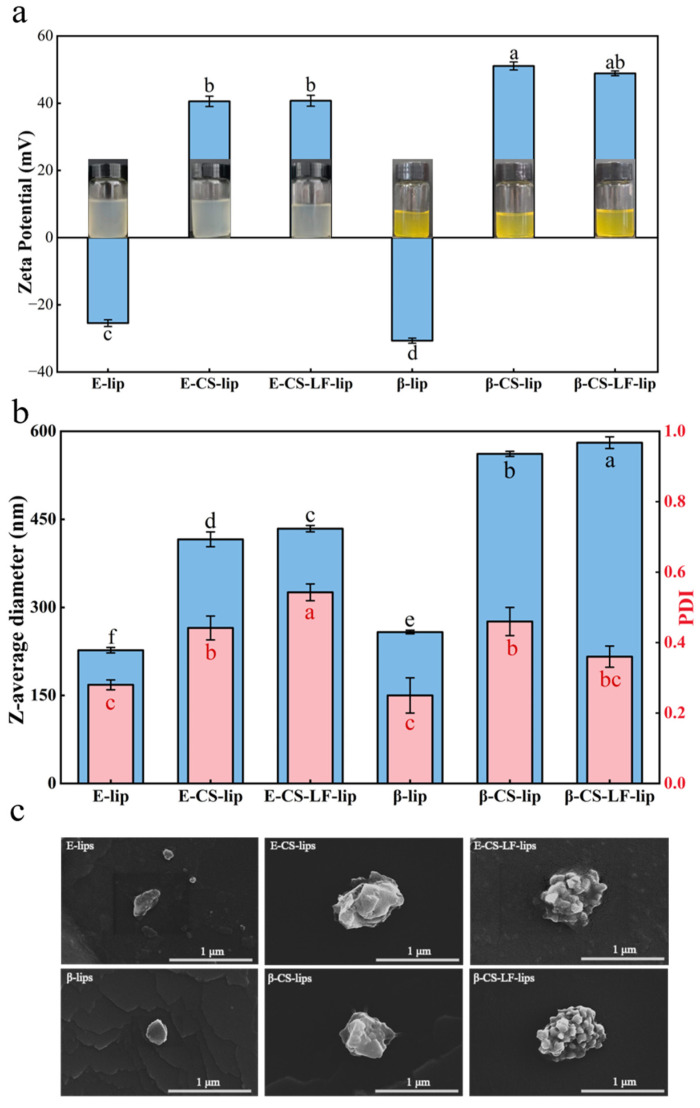
Particle size and optical photographs (**a**), zeta potential and PDI (**b**), SEM images (**c**) of E-lips, E-CS-lips, E-CS-LF-lips, β-lips, β-CS-lips, β-CS-LF-lips. Different letters indicate significant differences between Lips with different degrees of modification (*p* < 0.05).

**Figure 2 foods-14-00968-f002:**
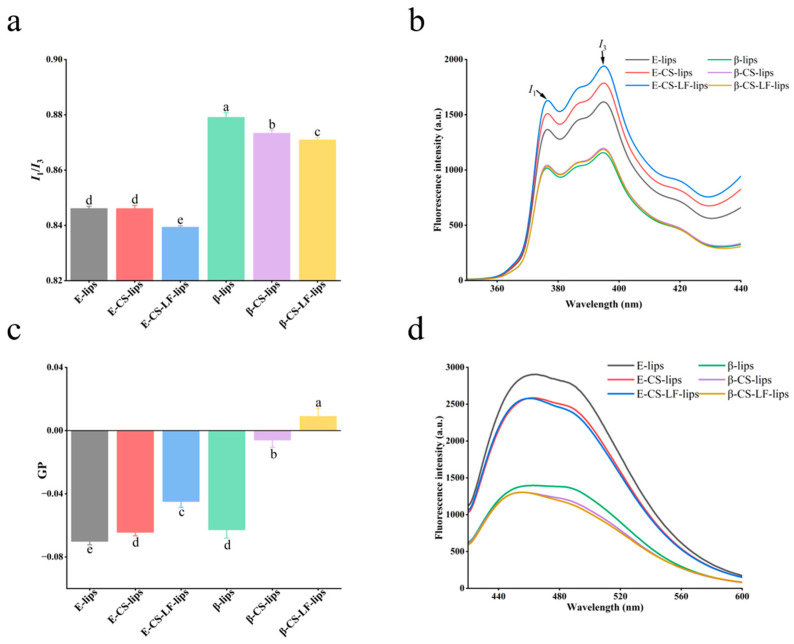
Membrane properties of Lips. Micropolarity (**a**), pyrene spectrograms (**b**), GP values (**c**) and fluorescence emission spectra after incubation with Laudan (**d**) of E-lips, E-CS-lips, E-CS-LF-lips, β-lips, β-CS-lips, β-CS-LF-lips, and β-CS-LF-lips. Different letters indicate significant differences between Lips with different degrees of modification (*p* < 0.05).

**Figure 3 foods-14-00968-f003:**
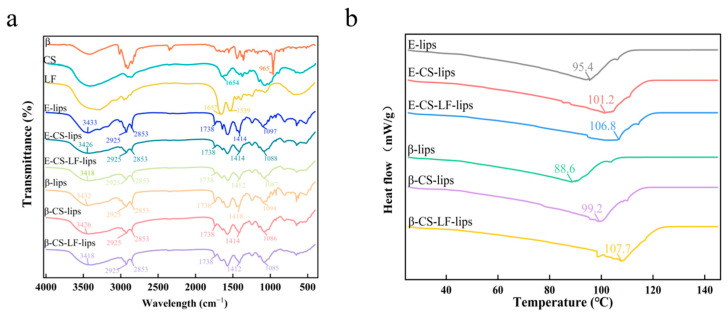
FTIR spectra of βC, CS, LF and E-lips, E-CS-lips, E-CS-LF-lips, β-lips, β-CS-lips, β-CS-LF-lips (**a**), and DSC spectra of E-lips, E-CS-lips, E-CS-LF-lips, β-lips, β-CS-lips, β-CS-LF-lips (**b**).

**Figure 4 foods-14-00968-f004:**
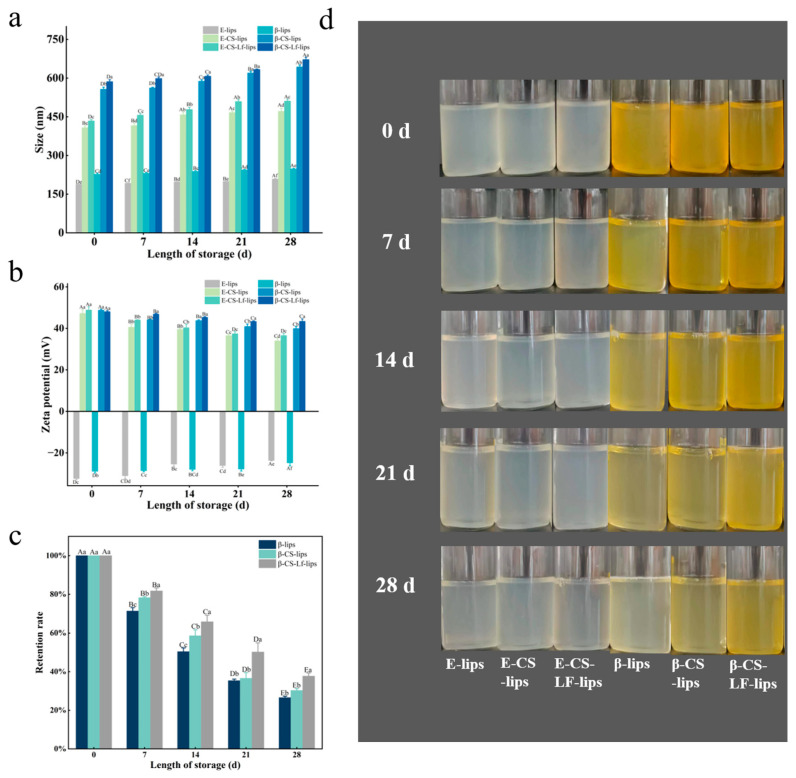
Storage stability of Lips. Particle size (**a**), potential (**b**), βC retention (**c**), and optical photographs (**d**) of E-lips, E-CS-lips, E-CS-LF-lips, β-lips, β-CS-lips, and β-CS-LF-lips during 28 days of storage at 4 °C. Different uppercase letters indicate significant differences between Lips with different storage times (*p* < 0.05), and different lowercase letters indicate significant differences between Lips under different degrees of modification (*p* < 0.05).

**Figure 5 foods-14-00968-f005:**
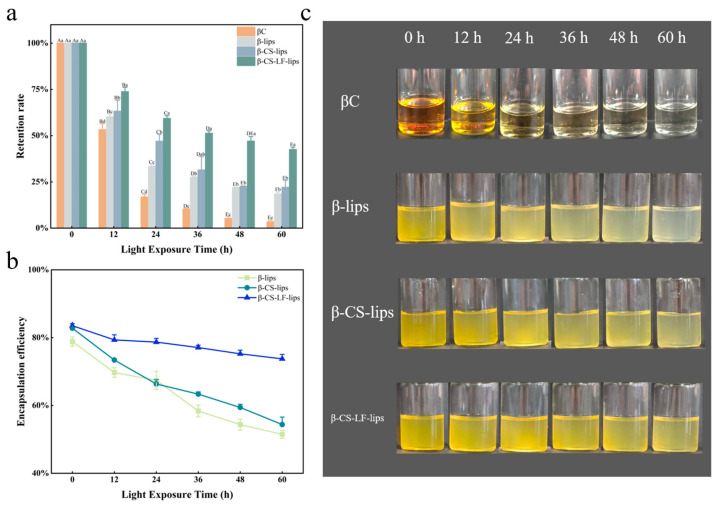
Photo stability of Lips. RR (**a**), EE (**b**) and optical photographs (**c**) of βC, β-lips, β-CS-lips, β-CS-LF-lips during UV illumination from 0 to 60 h. Different uppercase letters indicate significant differences between Lips with different UV illumination times (*p* < 0.05), and different lowercase letters indicate significant differences between Lips with different degrees of modification (*p* < 0.05).

**Figure 6 foods-14-00968-f006:**
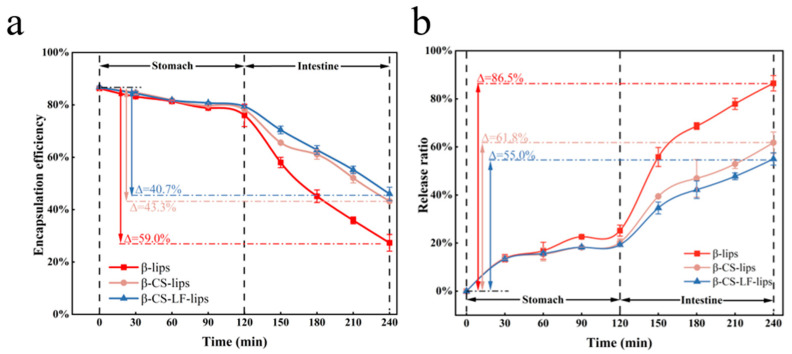
Lips digestion stability. EE (**a**), βC release curves (**b**) of βC, β-lips, β-CS-lips, and β-CS-LF-lips during digestion (SGF, SIF).

**Figure 7 foods-14-00968-f007:**
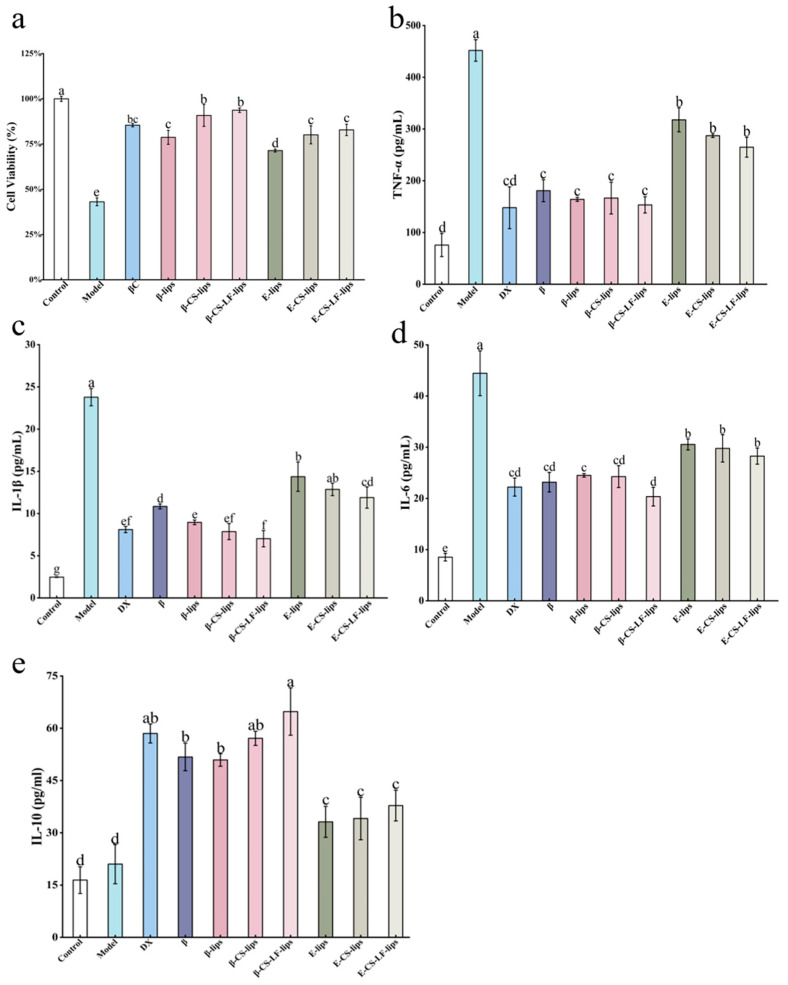
Liposomal anti-inflammatory property studies. The effects of βC, DX, E-lips, E-CS-lips, E-CS-LF-lips, β-lips, β-CS-lips, and β-CS-LF-lips on LPS-induced BV-2 cell viability (**a**), TNF-α (**b**), IL-1β (**c**), IL-6 (**d**), and IL-10 (**e**) levels. Different letters indicate significant differences between groups (*p* < 0.05).

**Table 1 foods-14-00968-t001:** The effects of different concentrations of βC on particle size, potential, PDI, and EE of Lip systems.

βC Load	Z-Average Diameter (nm)	Zeta Potential (mV)	PDI	EE
E-lips	167 ± 8 ^c^	−25 ± 1 ^a^	0.20 ± 0.03 ^c^	-
0.5%	231 ± 5 ^b^	−30.6 ± 0.3 ^b^	0.24 ± 0.04 ^b^	77.21 ± 2.22% ^b^
1.0%	258 ± 3 ^ab^	−30.7 ± 0.8 ^b^	0.25 ± 0.05 ^b^	84.91 ± 0.50% ^a^
1.5%	268 ± 6 ^a^	−31.9 ± 0.3 ^b^	0.28 ± 0.04 ^a^	70.39 ± 1.99% ^c^
2.0%	267 ± 7 ^a^	−30.5 ± 0.7 ^b^	0.25 ± 0.02 ^b^	68.95 ± 5.14% ^c^

Different letters mean the significance (*p* < 0.05). “-” represents that the EE of βC could not be determined by E-lips.

**Table 2 foods-14-00968-t002:** The effects of different concentrations of CS on particle size, potential, PDI, and EE of Lip systems.

CS Concentration	Z-Average Diameter (nm)	Zeta Potential (mV)	PDI	EE
0%	259 ± 3 ^f^	−33.0 ± 0.9 ^e^	0.27 ± 0.01 ^e^	77.21 ± 2.22% ^b^
0.5%	483 ± 5 ^e^	44.7 ± 0.4 ^d^	0.52 ± 0.01 ^bc^	76.98 ± 2.03% ^b^
0.6%	508 ± 8 ^d^	47.4 ± 0.4 ^c^	0.55 ± 0.01 ^ab^	80.71 ± 0.96% ^a^
0.7%	533 ± 13 ^c^	49.1 ± 1.2 ^bc^	0.44 ± 0.01 ^c^	81.53 ± 0.07% ^a^
0.8%	562 ± 4 ^b^	51.1 ± 1.2 ^a^	0.46 ± 0.04 ^cd^	81.85 ± 0.30% ^a^
0.9%	565 ± 12 ^b^	51.0 ± 1.0 ^ab^	0.53 ± 0.00 ^ab^	82.74 ± 0.92% ^a^
1.0%	818 ± 6 ^a^	50.2 ± 1.3 ^ab^	0.59 ± 0.08 ^a^	82.21 ± 1.14% ^a^

Different letters mean the significance (*p* < 0.05).

**Table 3 foods-14-00968-t003:** The effects of different concentrations of LF on particle size, potential, PDI, and EE of Lip systems.

LF Concentration	Z-Average Diameter (nm)	Zeta Potential (mV)	PDI	EE
0%	555 ± 13 ^b^	51.0 ± 0.3 ^a^	0.47 ± 0.03 ^b^	79.92 ± 0.78% ^c^
0.125%	552 ± 16 ^b^	49.9 ± 0.1 ^ab^	0.46 ± 0.03 ^b^	81.05 ± 0.48% ^bc^
0.25%	566 ± 6 ^a^	49.2 ± 1.3 ^b^	0.45 ± 0.02 ^b^	80.92 ± 0.53% ^bc^
0.375%	581 ± 10 ^a^	48.9 ± 0.7 ^b^	0.36 ± 0.03 ^c^	82.10 ± 0.72% ^ab^
0.5%	574 ± 10 ^a^	48.4 ± 1.2 ^b^	0.49 ± 0.01 ^ab^	83.10 ± 1.17% ^a^
0.1%	571 ± 4 ^a^	49.2 ± 1.2 ^b^	0.52 ± 0.04 ^a^	82.72 ± 0.36% ^a^

Different letters mean the significance (*p* < 0.05).

## Data Availability

The original contributions presented in this study are included in the article/Appendix A. Further inquiries can be directed to the corresponding author.

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
