# Peer review of "Stabilization of β-Carotene Liposomes with Chitosan–Lactoferrin Coating System: Vesicle Properties and Anti-Inflammatory In Vitro Studies"

_foods, 2025, doi:10.3390/foods14060968_

Round 1

Reviewer 1 Report

Comments and Suggestions for Authors

Line 98-103, Please provide more details on the lipid composition of fish head phospholipids, including their purity and composition profile.
Please, provide the final composition of liposomes in mg/mL or weight ratios for clarity.
Please indicate precise details of mechanical stirring or homogenization (e.g., speed, time), as these can influence particle size.
Line 118, Please clarify the basis for selecting β-carotene concentrations (0-2%) and how they compare to bioavailable dietary levels.
Line 257: The manuscript states that each experiment was done in triplicate (n=3), and significance is indicated. Please clarify the exact posthoc test used following ANOVA (e.g., Tukey’s, Duncan’s, LSD).
Line 346 Please elaborate on how lactoferrin incorporation affects membrane rigidity at a molecular level.
Line 394: Please provide more detail regarding the potential protein-polysaccharide complexation or self-assembly processes beyond the FTIR data about hydrogen bonding and electrostatic interactions. 
Line 468: Please expand on how electrostatic stabilization from chitosan prevents aggregation over 28 days.
Line 483 - Please discuss why β-carotene retention improved from 26.5% to 37.7% and how this compares to existing encapsulation techniques.
Line 490: any detais regarding preventing the beta-carotene oxidation 
Line 504-509, Please provide a possible mechanism for how UV exposure affects liposome degradation and compare with previous studies.
Line 540 - Please explore how bile salts and digestive enzymes interact with co-modified liposomes, potentially affecting bioavailability.
The improved anti-inflammatory effect is partially attributed to enhanced cellular uptake, especially with LF-labeled liposomes. While the in vitro data on reduced inflammatory markers are convincing, additional direct evidence of internalization (e.g., fluorescence imaging, flow cytometry) could further substantiate higher uptake by BV-2 cells. If not available, clarifying that it remains a hypothesis based on previous literature would be helpful.
•Line 591-600, Please describe whether β-carotene’s anti-inflammatory effects were dose-dependent and if further optimization is needed.
•Line 621-630, Please discuss how lactoferrin modification enhances cellular uptake of liposomes, leading to improved bioactivity.

4. Disucssion section is a list of results, please make a proper conclusion. 

Significant digits corrections: 

Table 1 
Please correct the following values: 
βC Load    Z-average diameter (nm)
1.0%    258.0 ± 3.1
1.5%    267.5 ± 6.2
2.0%    267.2 ± 7.2

Table 2 
βC Load    Z-average diameter (nm)
0.6%    508 ± 8
0.7%    533 ± 13
0.9%    565 ± 12
1.0%    818 ± 6

Table 3 
LF Concentration    Z-average diameter (nm) 
0%    555 ± 13
0.125%    552 ± 16
0.5%    574 ± 10

Author Response

We are very grateful for the constructive and insightful comments from the anonymous reviewers. Our point-to-point responses to their comments have been submitted as a Word file. Please see the attachment.

Sincerely yours,

Shuxin Gao

Reviewer 2 Report

Comments and Suggestions for Authors

I have reviewed the manuscript titled: "Stabilization of β-carotene liposomes with chitosan-lactoferrin coating system: studies on vesicle properties and anti-inflammatory in vitro." The study presents a well-structured investigation into the stabilization of β-carotene-loaded liposomes through co-modification with chitosan and lactoferrin. However, to enhance clarity and provide additional context for readers, I have outlined several suggestions for improvement.

Comments by Section

Abstract

  1. Clearly state the main hypothesis driving this study.
  2. Provide numerical values to support claims related to changes in particle size and zeta potential.
  3. Clarify whether the observed reduction in inflammation is primarily due to enhanced β-carotene stability or direct interactions with cells.

Introduction

  1. The instability of liposomes is described, but previous strategies for improving their stability are not discussed. Including this background would strengthen the justification for the study.
  2. Lactoferrin is introduced as a component of the coating system, but its specific advantages for this application are not explained. Elaborating on this would enhance the reader's understanding.

Materials and Methods

The methodology is appropriately described. No major concerns in this section.

Results and Discussion

  1. Particle size and zeta potential values are reported, but their implications for liposome stability are not sufficiently explained. A more detailed discussion is needed.
  2. The photostability of liposomes is analyzed in terms of β-carotene degradation, but it is unclear whether stabilization is due to a physical barrier effect or chemical interactions with chitosan/lactoferrin. This should be clarified.
  3. The anti-inflammatory effects are discussed, but it remains unclear whether they stem from enhanced β-carotene stability or direct interactions with cell membranes. This distinction should be addressed.

Conclusion

The conclusion is adequate.

Comments on the Quality of English Language

No comments

Author Response

(The authors gave the same response as above.)
